# Analyses of Swallowing Function and Its Related Factors in Community-Dwelling Elderly Patients: A Case-Control Study

**DOI:** 10.3390/jcm10153437

**Published:** 2021-08-02

**Authors:** Yoichiro Ogino, Hiroki Suzuki, Yasunori Ayukawa, Akio Jinnouchi, Kiyoshi Koyano

**Affiliations:** 1Section of Fixed Prosthodontics, Division of Oral Rehabilitation, Faculty of Dental Science, Kyushu University, Fukuoka 812-8582, Japan; ayukawa@dent.kyushu-u.ac.jp; 2Section of Implant and Rehabilitative Dentistry, Division of Oral Rehabilitation, Faculty of Dental Science, Kyushu University, Fukuoka 812-8582, Japan; zookey456odpmf@gmail.com; 3Department of Dentistry, Inouekai Medical Corporation Sasaguri Hospital, Sasaguri Town, Kasuya County, Fukuoka 811-2413, Japan; gin741219@icloud.com; 4Division of Advanced Dental Devices and Therapeutics, Faculty of Dental Science, Kyushu University, Fukuoka 812-8582, Japan; koyano@dent.kyushu-u.ac.jp

**Keywords:** swallowing function, oral hypofunction, community-dwelling elderly patients, tongue–lip motor function, maximum tongue pressure

## Abstract

This retrospective case-control study evaluated the prevalence of declined swallowing function and the association with oral functions and gender in community-dwelling elderly patients. Their profiles, the results of swallowing function (Eating Assessment Tool: EAT-10) and other oral functions (oral dryness, maximum occlusal force (MOF), tongue–lip motor function (oral diadochokinesis: ODK), maximum tongue pressure (MTP) and masticatory performance (MP)) were extracted for analyses. The patients were categorized into three groups according to EAT-10 score (Group 1: 0, Group 2: 1 and 2, Group 3: ≥3). In total, 242 patients were enrolled and 46 of them (19.0%) were categorized into declined swallowing function (Group 3). In two-group comparisons (Group 1, 2 versus Group 3), significant differences were identified in age and the number of remaining teeth, but they were not identified in three-group comparisons. The patients with declined swallowing function (Group 3) had significantly lower function in ODK and MTP. Multiple logistic regression analyses identified that declined swallowing function was independently associated with declined functions in ODK /ka/ (OR: 5.31, 95% CI: 1.03–27.23, *p* = 0.04) and in MTP (OR: 2.74, 95% CI: 1.12–6.66, *p* = 0.03). This study could confirm the critical role of tongue functions in swallowing in community-dwelling elderly patients.

## 1. Introduction

One of the health issues in the elderly is dysphagia [1,2]. Dysphagia, defined as a feeling that food sticks in throat or chest [3], is caused by aging, the decline in swallowing-related muscle functions including tongue, xerostomia, physical frailty, sarcopenia and comorbidities [1,2,3,4,5,6,7]. Furthermore, dysphagia often leads to malnutrition, aspiration pneumonia, morbidity and mortality [1,2,3,4,5,6,7,8]. Early recognition and diagnosis of this chronic symptom in the elderly is an urgent issue.

The swallowing process consists of three sequential phases; oral, pharyngeal, and esophageal phases [9]. This process requires a coordination of neuromuscular activity. Prior to swallowing, foods must be chewed and be mixed with saliva to form a bolus. The teeth or well-functioning dental prostheses, a sensory-motor activity by masticatory muscles and tongue, and salivary secretion play a crucial role in mastication [10,11,12,13]. The bolus is placed on tongue surface and is transferred to the pharynx by the elevation and the contraction of tongue and soft palate (oral phase). This process is a voluntary process. Sequentially, the bolus moves through the oropharynx (pharyngeal phase) and the esophagus (esophageal phase) into the stomach, which are involuntary processes [9,10,11]. Anatomical or physiologic disorders related to these processes result in dysphagia [14]. Especially, the poor oral functions might be directly relevant to oral phase dysphagia. However, oral functions in dysphagia patients have not been evaluated clearly.

The concept of “oral hypofunction” was introduced in 2018 [15]. This position paper proposed diagnostic criteria in seven oral signs or symptoms, which included a decline in swallowing function (dysphagia). Recent studies evaluated the oral functions and the association between oral functions and systemic conditions based on this concept [16,17,18,19,20,21]. It means that we can recognize that these oral functions are regarded as the typical oral functions.

The aims of this study were to evaluate the prevalence ratio of declined swallowing function in community-dwelling elderly patients and to compare their oral functions between the patients with and without declined swallowing function. The null hypothesis of this study was that there was no significant difference in oral functions between the patients with and without declined swallowing function and no independent factors are identified against swallowing function.

## 2. Materials and Methods

### 2.1. Ethical Approval and Study Population

This retrospective case-control study was approved by the institutional ethics committee (Approval Number #19). This study was conducted in accordance with the ethical principle of Helsinki Declaration. The community-dwelling elderly patients (65 years or older) who visited our institutional dental clinic for a regular check-up from April 2018 to March 2019 could be the candidates of this study. They had completed their prosthetic treatment and visited for the regular teeth cleaning and/or the maintenance of periodontal tissue and their prostheses. The inclusion criteria were as follows: (1) the patients who had been measured their oral functions described below after the completion of prosthetic treatment, (2) the patients who could use their prostheses without any problems, and (3) the patients where edentulous and/or partially edentulous sites were rehabilitated with fixed partial prostheses and/or removable dental prostheses. The following patients were excluded: (1) the patients who had ongoing dental treatment except for a regular check-up, (2) the patients who had the presence and the history of radiotherapy for head and neck cancer, and (3) the patients who had the presence and the history of the disease caused declined swallowing function such as stroke, head injury, tongue and salivary gland disease, or dementia.

At first, the patient profiles (age, gender, the number of remaining teeth and body mass index (BMI)) were extracted from the medical chart. Next, the enrolled patients were divided into three groups according to the score of Eating Assessment Tool (EAT-10; a self-administered questionnaire for swallowing) [15,16,18,20,21,22]: (1) EAT-10 score: 0 (Group 1), (2) EAT-10 score: 1 and 2 (Group 2), and (3) EAT-10 score ≥ 3 (Group 3). The assessment criterion for declined swallowing function (dysphagia) was EAT-10 score ≥ 3 (Group 3) [15,22].

### 2.2. Measurement of Oral Functions

Oral functions based on the concept of “oral hypofunction” had been measured [15]. Oral functions used for the analyses were as follows.

#### 2.2.1. Oral Dryness

Oral dryness was assess using an oral moisture checker (Mucus, Life Co., Ltd., Saitama, Japan) [15,16,17,18,20,21]. The sensor of this checker was pressed against tongue for a few seconds and the measurement value was presented [15]. The threshold value was 27 [15].

#### 2.2.2. Maximum Occlusal Force (MOF)

MOF was calculated using pressure indicating film (Dental Prescale II, GC, Tokyo, Japan) [15,16,17,19]. The patients were asked to clench the sheet in intercuspal position for 3 s and the sheet was analyzed using the software (Bite Force Analyzer, GC, Tokyo, Japan) to calculate MOF. The threshold value was 500 N [15].

#### 2.2.3. Tongue–Lip Motor Function (Oral Diadochokinesis: ODK)

ODK has been used to define tongue–lip motor function [15,16,17,18,20,21]. The patients were asked to utter the syllables /pa/, /ta/, or /ka/ as many as possible for 5 s. These sounds were recorded using a measurement device (Kenko-kun; Takei Scientific Instruments Co., Ltd., Niigata, Japan). This device could calculate the number of sounds per second after 5 seconds’ recording. In general, /pa/, /ta/ and /ka/ are used to evaluate motor function of lip, anterior region of the tongue and posterior region of the tongue respectively as a previous study showed [15]. The threshold value was 6 times per second [15].

#### 2.2.4. Maximum Tongue Pressure (MTP)

MTP was measured using a specific measuring device with a balloon of the probe (JM-TPM; JMS Co., Ltd., Hiroshima, Japan) [15,16,17,18,19,20,21]. The patients were instructed to compress the balloon on the tongue to anterior palate for 7 s. The device showed MTP and the average value was recorded after 3 measurements. The threshold value was 30 kPa [15].

#### 2.2.5. Masticatory Performance (MP)

MP was measured as the previous studies evaluated [15,16,17,19]. The patients were instructed to chew 2 g of gummy jelly voluntary for 20 s without swallowing. The crushed gummy jelly was spit out with saliva and 10 mL rinsing water. Glucose concentration from crushed gummy jelly was measured using a measuring device (Gluco Sensor GS-II, GC. Tokyo, Japan). The threshold value was 100 mg/dL [15].

### 2.3. Statistical Analyses

The numeric data including patient profiles (age, number of remaining teeth and BMI) and the measurement values of oral functions were presented as medians with interquartile ranges (IQRs). The comparisons between normal swallowing function groups (Group 1 and 2) and a declined swallowing function group (Group 3) were also evaluated by a Mann–Whitney U test (two-group comparisons). Three-group comparisons among Groups 1, 2, and 3 were performed via using the Steel–Dwass test after Kruskal–Wallis test. In addition, a multiple logistic regression analysis was also performed to examine whether declined swallowing function was independently associated with other oral functions and gender, and age- and the number of remaining teeth-adjusted odds ratios (ORs) with 95% confidence intervals (95% CIs) were also calculated. For all statistical analyses, JMP Pro 16 (SAS Institute Inc., Cary, NC, USA) was used and *p*-values less than 0.05 were considered statistically significant.

## 3. Results

### 3.1. Patient Profiles

Patient profiles were described in Table 1. In total, the enrolled patients were 242 community-dwelling elderly people (male and female: 89 and 153). The patients who were rehabilitated with removable dental prostheses (removable partial dentures and/or complete dentures) were 190 patients and 52 patients were natural teeth dental arches or were rehabilitated with fixed partial dentures (no removable dental prostheses). No patients were rehabilitated with implant prostheses. Oral functions in the patients with removable dental prostheses were measured with their dentures. The results of EAT-10 demonstrated that 46 patients (19.0 %, male and female: 23 (25.8%) and 23 (15.0%)) were defined as the patients with declined swallowing function. The statistical analyses in two-group comparisons revealed the significant differences in age (*p* < 0.05, Mann–Whitney U test) and the number of remaining teeth (*p* < 0.05, Mann–Whitney U test) between Group 1 and 2 versus Group 3, although no significant difference in BMI was identified. However, multiple statistical analyses identified no significant differences among three groups in these three categories (*p* > 0.05, Steel–Dwass test after Kruskal–Wallis test).

### 3.2. Comparisons of Oral Functions

Oral functions were also compared between Groups 1 and 2 versus Group 3 (two-group comparisons, normal swallowing function groups and a declined swallowing function group), and among three groups (three-group comparisons) (Table 2). In two-group comparisons, the patients who belonged to Group 3 showed the significant lower values in all ODKs and MTP compared to Group 1 and 2 patients (*p* < 0.01, except for ODK /pa/ sound (*p* < 0.05), Mann–Whitney U test). Furthermore, three-group comparisons demonstrated that there were significant differences in ODK each sound between Group 1 and Group 3 (/pa/ and /ta/; *p* < 0.05, /ka/; *p* < 0.01, Steel–Dwass test after the Kruskal–Wallis test), and in MTP between Group 1 and Group 3 (*p* < 0.01, Steel–Dwass test after Kruskal–Wallis test), and between Group 1and Group 2 (*p* < 0.05, Steel–Dwass test after Kruskal–Wallis test).

### 3.3. Odds Ratios (ORs) and a Multiple Logistic Regression Analysis

The results of multiple logistic regression analyses including calculations of ORs were shown in Table 3. The model summary revealed that this model was statistically significant (*p* < 0.01). This analysis identified that declined swallowing function (EAT-10 score ≥ 3) was independently associated with declined functions in ODK /ka/ (OR: 5.31, 95% CI: 1.03–27.23, *p* = 0.04) and in MTP (OR: 2.74, 95% CI: 1.12–6.66, *p* = 0.03).

## 4. Discussion

As the previous studies reported, declined swallowing function (dysphagia) has been known as a multifactorial disorder [1,2,3,4,5,6,7]. A recent study based on a self-report questionnaire with a frailty checklist presented that dysphagia was independently associated with oral, physical, cognitive and psychological frailty in community-dwelling elderly people [23]. Swallowing is a complex behavior including the reflexive and voluntary actions, and can be assumed as a result of the coordination of several functions including perioral muscles, tongue and neural network [9,10]. Hence, this study was conducted as a retrospective case-control study to compare the measurement values of oral functions between community-dwelling elderly patients with and without declined swallowing function. However, this study also aimed to identify the factors independently associated with declined swallowing function.

Since the concept of “oral hypofunction” has been introduced, typical oral functions become evident. The oral functions evaluated in this study were selected based on this concept. The present study clearly showed that the significant differences in tongue–lip motor function (ODK) and MTP were observed between the elderly patients with and without declined swallowing function, and above all it is suggested that motor function of posterior region of tongue (ODK /ka/) and MTP play a crucial role in swallowing function. These findings clearly showed the importance of tongue functions in swallowing. The previous studies indicated a crucial role of tongue pressure or MTP in swallowing [24,25,26,27]. The key role of tongue–lip motor function in swallowing was suggested in several studies [28,29] and the studies also showed the prevalence of the elderly people who showed declined swallowing function and lower ODK [30,31,32]. Regarding tongue–lip motor function (ODK), clear evidence for the association of ODK with swallowing function is still insufficient, because there have been few studies that evaluated swallowing function using three of tongue–lip motor functions proposed by Minakuchi et al. On this point, this study demonstrated the significance of motor function of posterior region of tongue (ODK /ka/) as a result of three kinds of ODK evaluations. Interestingly, MTP in this study was measured by the elevation of the tongue. This motion works for the transfer of bolus to the back of the mouth and is followed by the protrusion of the tongue to squeeze the bolus back along the palate and into the pharynx [9,10]. The former motion is similar to /ta/ pronunciation and the latter is similar to /ka/ pronunciation. This study confirmed the significance of these motions, and the motor function of posterior region of tongue or the protrusion of the tongue measured by ODH /ka/, in particular, can be one of the crucial factors in swallowing function. We believe that these findings can recommend MTP and ODK evaluation for the patients with declined swallowing function as screening factors, and the rehabilitation of both functions might contribute to its improvement. As the previous studies showed, the tongue strengthening exercise is helpful in improving swallowing function [33,34,35,36], although further well-designed studies will be required.

In this study, there were no differences in the other oral functions between the patients with and without declined swallowing function. However, the previous studies suggested the association of each oral functions with swallowing. Although oral dryness or xerostomia could be a risk factor of declined swallowing function, the previous studies suggested that it depended on the severity of oral dryness [37,38]. The subjects in this study were relatively better condition as presented in Table 2, resulting that we could not identify oral dryness as a significant factor. Regarding MOF and MP, it is important to note that the previous studies showed the significant correlation between these two factors [12,39,40,41,42]. In addition, several studies showed the correlation between MP and MTP [12,13,42,43], and between MOF and MTP [44]. Although this study suggested more critical role of MTP in swallowing function compared to these factors, we need to recognize the correlations between these factors and swallowing function. This means that relative effects of them on swallowing function should be noted for the rehabilitation

Although we believed that this study has revealed novel and critical findings, we need to describe the limitation of this study, which is the characteristics of the subjects in this study. The subjects of this study were community-dwelling elderly patients who visited for a regular check-up after the completion of the dental treatment. As described in inclusion and exclusion criteria, all of the subjects might be defined as healthy elderly population. However, we did not follow the detailed systemic conditions such as the history of systemic diseases and medications, which could be confounding factors. In addition, it is not clear the ratios of elderly people who can visit dental clinics. This means that the definition of “community-dwelling elderly patients” is not same as “community-dwelling elderly people”. According to this limitation, we decided that the title of this study included “community-dwelling elderly patients”, not “community-dwelling elderly people”. In addition, the subjects were limited to the patients due to the retrospective study and the sample size was not enough. However, we tried to calculate the sample size based on the previous studies [21,23,45], and the prevalence of these factors were wide varieties and it was difficult to calculate the sample size. The data analyses regarding the differences between genders and specific findings in each gender were not conducted due to the limitation of sample size. However, multiple logistic regression analysis revealed that *p*-value in gender was 0.08 and suggested the effect of gender on swallowing. We hope to have further studies that include a wider range of population, although we believe the result of this study was beneficial to understand the etiology of declined swallowing function.

## 5. Conclusions

Within the limitation of this study, the present analyses suggested 19.0 % of the subjects (male: 25.8% and female: 15.0%) showed the declined swallowing function. Analyses of oral functions revealed that MTP and tongue–lip motor function in the patients without the declined swallowing function were significantly better than the patients with the declined swallowing function. Multiple logistic regression analysis revealed that the motor function of posterior region of tongue (ODK /ka/) and MTP were significant independent factors in swallowing function. However, the subjects of this study were limited to the patients who could visit dental clinics for a regular check-up and more subjects who have various backgrounds must be enrolled to evaluate the real swallowing etiology in community-dwelling elderly people.

## Figures and Tables

**Table 1 jcm-10-03437-t001:** Profiles of enrolled patients.

	All Patients	Groups 1 and 2 (EAT-10: 0–2)	Group 1 (EAT-10: 0)	Group 2 (EAT-10: 1, 2)	Group 3 (EAT-10 ≥ 3)
Gender (male:female)	89:153	66:130	48:90	18:40	23:23
Age	78 (71–84)	77 * (71–83)	77 (71–82)	78 (74–84)	82 (71–87)
Number of remaining teeth	18 (7–24)	19 * (8–25)	18 (7–25)	20 (9–26)	11.5 (4.75–21.25)
BMI	22.3 (19.9–24.6)	22.5 (20.3–24.9)	22.4 (20.4–24.6)	22.5 (19.5–25.2)	21.4 (18.4–24.2)

EAT-10: Eating Assessment Tool-10. The values were described as median and interquartile range. * *p* < 0.05, vs. Group 3, Mann–Whitney U test (two-group comparison). No significant differences among Groups 1, 2 and 3, Steel–Dwass test after Kruskal–Wallis test (three-group comparison).

**Table 2 jcm-10-03437-t002:** Measurement values (medians and interquartile ranges) of oral functions in each group.

	All Patients	Groups 1 and 2 (EAT-10: 0–2)	Group 1 (EAT-10: 0)	Group 2 (EAT-10: 1, 2)	Group 3 (EAT-10 ≥ 3)
Oral dryness	27.1 (24.6–29.4)	27.0 (24.4–29.4)	27.0 (24.4–29.5)	26.9 (24.5–29.0)	27.2 (26.0–29.6)
MOF	406.7 (229.4–663.6)	422.2 (237.1–683.2)	419.0 (229.3–659.6)	470.3 (254.5–711.1)	328.2 (215.6–581.4)
ODK /pa/	5.6 (4.4–6.2)	5.8 * (4.8–6.2)	5.8 (4.8–6.2)	5.6 (4.5–6.1)	4.9 ^#^ (3.8–6.1)
ODK /ta/	5.4 (4.6–6.2)	5.4 ** (4.6–6.2)	5.5 (4.8–6.2)	5.3 (4.3–6.2)	5.0 ^#^ (4.1–5.8)
ODK /ka/	5.0 (4.0–5.8)	5.2 ** (4.2–5.8)	5.2 (4.4–5.8)	5.0 (4.1–5.9)	4.5 ^##^ (3.4–5.5)
MTP	25.7 (20.0–31.3)	27.3 ** (20.8–32.5)	28.0 (21.6–34.0)	24.3 ^$^ (17.1–29.3)	22.3 ^##^ (16.0–26.9)
MP	121.5 (82.0–161.0)	122.0 (83.0–162.8)	122.0 (83.0–163.5)	118.5 (81.5–164.0)	114.5 (76.5–151.5)

MOF: maximum occlusal force, ODK: oral diadochokinesis for tongue–lip motor function, MTP: maximum tongue pressure, MP: masticatory performance. * *p* < 0.05 and ** *p* < 0.05, vs. Group 3, Mann–Whitney U test (two-group comparison). ^#^
*p* < 0.05, ^##^
*p* < 0.01 and ^$^
*p* < 0.05, vs. Group 1 Steel–Dwass test after Kruskal–Wallis test (three-group comparison).

**Table 3 jcm-10-03437-t003:** Odds ratios and multiple logistic regression analysis.

	Odds Ratio	95% Confidence Interval	*p*-Value (Multiple Logistic Regression Analysis)
Oral dryness	0.71	0.36–1.41	0.33
MOF	1.88	0.83–4.28	0.13
ODK /pa/	0.99	0.37–2.65	0.98
ODK /ta/	0.83	0.28–2.47	0.74
ODK /ka/	5.31	1.03–27.23	0.04
MTP	2.74	1.12–6.66	0.03
MP	0.61	0.29–1.32	0.21
Gender	0.54	0.27–1.07	0.08

MOF: maximum occlusal force, ODK: oral diadochokinesis for tongue–lip motor function, MTP: maximum tongue pressure, MP: masticatory performance.

## Data Availability

The datasets used and/or analyzed during the current study are available from the corresponding author upon reasonable request.

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
