# Peer review of "Analyses of Swallowing Function and Its Related Factors in Community-Dwelling Elderly Patients: A Case-Control Study"

_jcm, 2021, doi:10.3390/jcm10153437_

Round 1
Reviewer 1 Report
This manuscript describes an interesting case-control study analysing the swallowing function and its related factors in a sample of 242 community-dwelling elderly patients (46 patients with declined swallowing function and 196 control patients). In my opinion, this is a well-developed work that provides comprehensive results showing a prevalence of 19.0 % of declined swallowing function. Furthermore, authors investigated the oral function of all patients analysing several related factors. Authors must first address a few considerations:
- Authors must ask for a proofreading of the manuscript by a native English-speaking specialist. Despite English is not my native language, I can detect many sentences that could be more readable after a professionally proofread. The best example of this observation can be found in the first sentence of the "conclusions" section (I am not able to understand its meaning well).
- The “abstract” is another example of poor writing. Furthermore, authors should state in the abstract the prevalence of declined swallowing function showed in their sample.
- Authors indicate in the “abstract” section “This … study evaluated the prevalence of declined swallowing function and the association with oral functions and gender in community-dwelling elderly patients”. However, in the main body of the manuscript the gender of the patients is not fully analysed. For example, according to my own calculations, the prevalence of declined swallowing function in their sample is different in males (25.8 %) and females (15.0 %). Authors should analysed the gender variable more extensively.
Author Response
We really appreciate your comments. Our opinions and revisions were as follows.
This manuscript describes an interesting case-control study analysing the swallowing function and its related factors in a sample of 242 community-dwelling elderly patients (46 patients with declined swallowing function and 196 control patients). In my opinion, this is a well-developed work that provides comprehensive results showing a prevalence of 19.0 % of declined swallowing function. Furthermore, authors investigated the oral function of all patients analysing several related factors. Authors must first address a few considerations:
- Authors must ask for a proofreading of the manuscript by a native English-speaking specialist. Despite English is not my native language, I can detect many sentences that could be more readable after a professionally proofread. The best example of this observation can be found in the first sentence of the "conclusions" section (I am not able to understand its meaning well).
The “abstract” is another example of poor writing. Furthermore, authors should state in the abstract the prevalence of declined swallowing function showed in their sample.
> We appreciate your comments. We got English revision by a native speaker and you can find the revisions in the text.
- Authors indicate in the “abstract” section “This … study evaluated the prevalence of declined swallowing function and the association with oral functions and gender in community-dwelling elderly patients”. However, in the main body of the manuscript the gender of the patients is not fully analysed. For example, according to my own calculations, the prevalence of declined swallowing function in their sample is different in males (25.8 %) and females (15.0 %). Authors should analysed the gender variable more extensively.
> We can totally agree your comments. When we analyzed the data, we considered the effect of gender on swallowing and the specific features in each gender. The reasons we gave up showing the gender analyses were limited sample sizes and complicated data. Especially, if we have shown the patient profiles and all oral functions in each gender, we would consider the readers would be confused. Of course, we analyzed the data in each gender population, and similar finding were observed. In addition, odds ratios were also calculated, and interestingly only MTP in male was the significant. We decided these analyses were favorable with a sufficient number of subjects. Alternatively, in multiple logistic regression analysis, we adopted gender as one of explanatory variables. The result was not significant in this sample size, but this finding suggested the interests in gender analyses. Following these reasons and results, we decided to show only the results of multiple logistic regression analysis. We added some comments in Discussion. We hope you understand this situation. However, if you need gender analyses in this study, we could show them.
Reviewer 2 Report
The paper itself is well written and documented, showing a great effort from the authors.
Statistical analysis is appropriate.
References are recent.
The topic is quite original and maybe of little clinical relevance.
I would make only the following few mentions:
The inclusion criteria should consider not only the number of remaining teeth but also the type of prosthodontic rehabilitation. Maybe subdividing the subject in fixed or removable prosthesis. Why patient with implant-supported prostheses have been excluded? Another exclusion criteria could be any pathology of the salivatory glands or tongue.
Otherwise, the article is good.
Author Response
The paper itself is well written and documented, showing a great effort from the authors. Statistical analysis is appropriate. References are recent. The topic is quite original and maybe of little clinical relevance. I would make only the following few mentions:
The inclusion criteria should consider not only the number of remaining teeth but also the type of prosthodontic rehabilitation. Maybe subdividing the subject in fixed or removable prosthesis. Why patient with implant-supported prostheses have been excluded? Another exclusion criteria could be any pathology of the salivatory glands or tongue.
Otherwise, the article is good.
>We really appreciate your comments and we’re so glad to read them. As you suggested, we added the comments about prosthetic conditions in “Results”. In addition, we made an error to describe about implant-supported prostheses. We did not find the patients who had been rehabilitated with implants and no data of these patients. The exclusion criteria were revised including the disease of tongue and salivary gland.
Round 2
Reviewer 1 Report
In my opinion the manuscript is now suitable for publication in the JCM.